# Graphene-Based Nanocomposites for Neural Tissue Engineering

**DOI:** 10.3390/molecules24040658

**Published:** 2019-02-13

**Authors:** Ho Pan Bei, Yuhe Yang, Qiang Zhang, Yu Tian, Xiaoming Luo, Mo Yang, Xin Zhao

**Affiliations:** 1Department of Biomedical Engineering, The Hong Kong Polytechnic University, Hung Hom, Kowloon 999077, Hong Kong, China; peter.bei@connect.polyu.hk (H.P.B.); yuhe.yang@u.nus.edu (Y.Y.); qiang666.zhang@connect.polyu.hk (Q.Z.); yu.xz.tian@polyu.edu.hk (Y.T.); 2Department of Mechanical Engineering, National University of Singapore (NUS), Singapore 117575, Singapore; 3Department of Preventive Medicine, School of Public Health, Chengdu Medical College, Chengdu 610500, China; lxm@cmc.edu.cn

**Keywords:** graphene, nanocomposites, neural tissue engineering

## Abstract

Graphene has made significant contributions to neural tissue engineering due to its electrical conductivity, biocompatibility, mechanical strength, and high surface area. However, it demonstrates a lack of biological and chemical cues. Also, it may cause potential damage to the host body, limiting its achievement of efficient construction of neural tissues. Recently, there has been an increasing number of studies showing that combining graphene with other materials to form nano-composites can provide exceptional platforms for both stimulating neural stem cell adhesion, proliferation, differentiation and neural regeneration. This suggests that graphene nanocomposites are greatly beneficial in neural regenerative medicine. In this mini review, we will discuss the application of graphene nanocomposites in neural tissue engineering and their limitations, through their effect on neural stem cell differentiation and constructs for neural regeneration.

## 1. Introduction

Peripheral nerve injury (PNI) has been widely reported to be an essential cause of incapability in the clinical setting. Patients suffering from this disease encounter many challenges including pain, sensory loss, impaired movement, and cold intolerance, which have a significant adverse impact on their daily life [1]. The most common therapeutic strategy for PNI is painstaking microsurgical mending by tensionless epineural sutures. Even with a substantially increased understanding of neuropathophysiology, there is still a lack of appropriate strategies for accelerated growth and complete recovery of axon during the neurorepair [2,3]. While neurons are the most vital part of both neural systems, they cannot perform mitosis, and supporting glial cells can only divide with limited capacity. In addition to slow/inability to heal, protease activity degenerates distal ends after a nerve is severed. Hence, the human neural system cannot regenerate by itself naturally and is prone to permanent damage due to injury and disease. Recently, increasing evidence has showed that neural tissue engineering is able to help neural cells recover using constructs such as scaffolds, sheets, and nanoparticles fabricated through bioengineering, which actively stimulates nerve regeneration through topological, physiochemical and biological cues by incorporation of biocompatible materials and therapeutic molecules such as growth factors [4,5]. In particular, flexible materials are favored for neural regeneration due to their ability to recapitulate and support intrinsic tissues.

Recent advances in tissue engineering and regenerative medicine have paved the way for the development of many strategies in peripheral nerve repair and various functional topographical scaffolds such as gels, films, electro spun fibers, and grooves which have been successfully demonstrated to support neuronal cellular attachment and migration [6]. Due to the speedy development of fabrication and functionalization methods, graphene and its related derivatives with their multiple, fascinating physi-chemical properties have opened numerous opportunities in biomedical areas [7,8,9,10]. Graphene is single layers of carbon atoms connected to each other in *sp*^2^ hybridization, forming a honeycomb pattern throughout the 2D plane [11]. Highly mobile carriers in graphene monolayers enable them to possess excellent electrical conductivity and zero bandgap like metals, which is unique in non-metal materials [12]. Notably, it has been demonstrated by many researchers that electrical stimulus is beneficial to neuronal regrowth [13,14]. Compared to other novel materials such as indium tin oxide and conductive polymers (Table 1), the excellent electrical conductivity, flexibility and mechanical strength of graphene-based materials allow them to excel at neural tissue engineering [8]. 

However, while pristine graphene can induce synaptogenesis and sustain neuronal growth, pure graphene sheets fail to recapitulate the intrinsic neural microenvironment in vivo due to its smoothness, chemical inertness and lack of biological cues [18,19,20]. It has been evidenced that nanostructures such as fibers, grooves, and ridges can promote cell attachment and proliferation of new tissues, while supplementing with growth factors greatly improves neural cell growth and alignment [21,22]. In addition, the strong π–π interaction characterizing the face-to-face stacking of aromatic systems on graphene surface are hypothesized to induce bacterial interactions with cell membranes, causing substantial damage. Currently, large scale production of pure graphene relies on chemical vapor deposition, which requires large amounts of energy for production leading to increased cost and environmental concerns [23]. Thus, derivatives of graphene have been modified by many researchers to form nanocomposites such as graphene oxide (GO) to improve the surface roughness and promote neural growth [21,24,25,26]. Platelets of graphene have also been infused with other bulk materials to strengthen overall mechanical and electrical properties [27]. For example, due to the high mobility of charge carriers in graphene, different parts of graphene constructs can be tailored to have a variable surface charge to interface with a neural network, which signals neurons to exhibit increased neurite length and total outgrowth [28]. Furthermore, graphene nanocomposites can be functionalized to load growth factors and/or drugs or offer ways to bond with biological molecules such as proteins, carbohydrates and nucleic acids [14,29,30]. These diverse forms of graphene nanocomposites and their high electrical capabilities as an electrode allows graphene to recapitulate and record the nervous system’s intricate signaling processes. 

In this mini review, we will first touch upon the structure and functions of graphene nanocomposites. The main focus will be on recent research advances in graphene-based nanocomposites for application in neural tissue engineering. Finally, we will provide a summary and perspectives for using graphene nanocomposites as the next generation of materials for neural tissue engineering.

## 2. Structure and Functionalization of Graphene-Based Nanocomposites

Graphene-based nanomaterials (GNs) can be subdivided according to the number of layers, sheet dimensions, surface chemical modification status, and defect density. The most widely applied ones are single-layer graphene, few-layered graphene, GO, as well as reduced graphene oxide (rGO), as shown in Figure 1. As the name indicates, single layer graphene is a separated single layer of carbon atoms arranged in a planar hexagonal lattice, and few-layered graphene is an ordered stacking of the single layer graphene. Although they have excellent electronic and mechanical property, the difficulties in the bulk production of defect-free material, solution suspending and isolation in the ambient atmosphere hinder their application in neural tissue engineering. GO is a single highly oxidized graphene sheet with oxygen-containing moieties including peripheral carboxylic acid (-COOH) [8], in-plane epoxide (-O-) and hydroxyl (-OH) groups [31]. The negative charged carboxylate groups on GO can increase the colloidal stability and hydrophilicity, making it more suitable for surface attachment, proliferation and differentiation of neurocytes. The uncharged epoxide and hydroxyl groups on the basal plane can conveniently induce some physical interactions and chemical surface reaction. Moreover, the non-functionalized areas of graphene on the basal planes remain repellant to water and allow for π–π interaction, so it has been widely suggested as an amphiphilic material [21]. However, considerable defect density caused by the presence of functional groups in GO can decrease its electrical conductivity and mechanical strength. rGO, a structural intermediate between graphene and GO with reduced oxygen content and enhanced optical and electrical properties [31], can be mass synthesized from GO by various reducing conditions. Due to its lower cost, easy fabrication, enhanced availability as well as better electrical conductivity, rGO is often used in place of pure graphene in neural tissue engineering [32]. In addition to the single physi-chemical property of the graphene-based nanomaterials, the forms of GNs (number of layers, lateral dimension, etc.) could alter their stiffness and flexibility and also r have an important effect on their interactions with cells and tissues. For example, Mu et al. reported an uptaking mechanism shift from clathrin-mediated endocytosis for small GO nanosheets (thickness 3.2 nm) to clathrin-mediated endocytosis and phagocytosis for large GO nanosheets (thickness 5.2 nm) [33]. In addition, the cyto- and geno-toxicity of the GNs are also highly relevant to the average lateral dimensions (ALD) as cell destructions can be observed with 11 nm ALD at low concentration of rGO, while the rGO sheets with 3.8 nm ALD presented a noticeable cytotoxicity effect only at high concentrations [34]. 

Inspired by their fascinating physical and chemical properties, the GNs have been widely applied to neural tissue engineering; however, some substantial problems have significantly hindered their application. For example, most of the above GNs have been demonstrated to possess an aggregation tendency, cytotoxicity and insufficient bioactivity in physiological environments because of the strong interaction among nano-sized graphenes, uncleared biocompatibility and their inherent bio-inert nature [35]. Thus, functionalization of the GNs is regarded as the practical solution to improve its capacity in a specific application, and this can be easily realized by covalent modification by forming amide and carbamate ester bonds with the carboxyl and hydroxyl groups, respectively. In addition, non-covalent modification through weak van der Waals forces, π electrons stacking, and electrostatic interaction has also been utilized to optimize the surface properties of graphene [36]. Obviously, GO is easier for functionalization because of the presence of the multi oxygen-containing functional groups on its surface as well as its good dispersion in water. GO has been covalently functionalized with some water-soluble polymers terminated with hydroxyl groups through esterification including polyethylene glycol [37,38], polyvinyl alcohol [39,40], amphiphilic copolymers [41,42], amino groups [43], and sulfonic groups [44]. Several GO covalently-functionalized nanocomposites are proven to have increased stability in physiological solutions and a significant reduction in toxic side effects. Besides, non-covalent functionalization of GNs is also widely reported in both hydrophilic and hydrophobic areas [45,46]. Generally, non-covalent functionalization is a more versatile strategy for GNs, which enhances the solubility or stability of GNs in aqueous condition by a surfactant or amphiphilic polymer. Absorbing and stabilizing the polymer on the graphene sheets in water is an efficient way to produce high-quality water stable/soluble GNs. Therefore, polymers and biomolecules (protein, DNA, etc.) can be used to decorate the GNs to extend its application in tissue engineering [47,48].

## 3. Applications of Graphene-Based Nanomaterials in Neural Tissue Engineering

### 3.1. Neuronal Differentiation of Sem Cells on Graphene-Based Nanomaterials

Utilizing graphene’s unique surface and electrical properties to mimic in vivo cues to enhance the differentiation of stem cells into neurons is an extremely attractive strategy for the treatment of neuronal injuries and diseases. Reports have shown that GN-based materials promote stem cells to differentiate into neurons, as well as improve their proliferation rate [18,49,50]. However, the exact mechanism of the effect of graphene on neural stem cell differentiation remains unknown to this day. Here we discuss different approaches that researchers have used to promote neuron differentiation from stem cells, as well as the future direction of related studies.

In one study, Akhavan et al. cultured neural stem cells (NSCs) on ginseng-reduced GO nanosheets to improve the material’s biocompatibility and capability for electron transfer [25]. Using ginseng extract as the substrate and Fe foil as the catalyst, GO sheets were reduced, forming thicker sheets of 2.9 nm with the attachment of ginsenosides, flavanones, and starch on the surface. The added molecules were shown to be beneficial to neuronal growth and differentiation. After 3 weeks of incubation, the overall cell density on ginseng-reduced GO is nearly ten times that of pure GO. Ginseng-reduced GO also exhibited six times more neurons than glial cells while the two cell types showed a similar ratio of pure GO. These results proved that the functionalization of biomolecules on graphene-based materials could provide useful chemical cues, thus improving graphene’s ability to aid the differentiation of NSCs into neurons. 

To investigate the effect of the physical cues of graphene composites on human neural stem cells (hNSCs), Solanki et al. functionalized laminated GO nanoparticles on charged glass to form aligned patterns, matching the extracellular matrix (ECM) protein patterns in the human body [51]. GO was chosen in place of pure graphene due to its better capability for laminin assembly. The experimental setups were positively charged glass with laminin, glass with a monolayer of positively charged NPs, positively charged glass with graphene coating and glass with both positive NPs and graphene. Starting from day 5, the axons of hNSCs began to align only on GO substrates but not on glass, forming highly aligned patterns with a variation of angles of only ±17.8°. Similar results can be reached with half the cell seeding density, confirming that axonal alignment of differentiating hNSCs was dependent on GO, not crowding of cells (Figure 2). To investigate the difference between GO only and GO with NPs, qPCR analyses on mRNA of neuronal marker TuJ1, GAP43, MAP2, and synapsin expression levels were performed. It was shown that the expression levels of neuronal markers on GO with NPs was significantly higher than naked GO. Results suggest that physical cues provided by silica NPs induce increased maturity in hNSCs.

However, the aforementioned studies were only able to capture the differentiation of NSCs on a 2D plane, which is very different from the three-dimensional (3D) ECM nanostructure. The signaling between cell-microenvironment interaction and attachment of cells onto a 3D dynamic network of macromolecules is essential to maintaining stem cell homeostasis [52]. In addition to allowing the cells to proliferate in a 3D microenvironment and directing them to a specific lineage, 3D GNs provide excellent electrical coupling between the material and neurons due to interconnected 3D neural networks, resulting in increased conductivity and electroactivity [53]. For instance, Shah et al. investigated the effect of GO concentration on 3D nanofibrous structures on differentiation of NSCs into oligodendrocytes. Electro spun fibers of 200–300 nm were fabricated using polycaprolactone (PCL) and collected on a glass substrate, then coated with GO of three different concentrations (0.1, 0.5 and 1.0 mg/mL). Same glass substrates were also coated with GO solutions without the nanofibers as control and were shown to have less adhered GO due to decreased surface area, suggesting a high affinity between PCL and GO. Results showed that substrates with nanofibers exhibited higher oligodendrocyte differentiation while lowering neuron and glial cell differentiation, with more pronounced expression of myelin basic protein at higher GO concentrations, which is beneficial for neural regeneration. PCL-GO showed over double the amount of oligodendrocyte marker Olig2 expression than PCL or GO only, demonstrating that the synergistic effect of the nanofibrous scaffold and GO conductivity on NSCs were able to complement each other in neural tissue engineering. 

### 3.2. Graphene-Based Nanomaterial Construct for Neural Regeneration

Other than promoting neuronal differentiation to replace damaged tissues, GNs can also directly stimulate neural regeneration as electrodes. Electrical stimulation is vital to promoting cell proliferation and mimicking in vivo neural networks as neural cells are interconnected via neurites in native tissues [54]. For example, Akhavan et al. utilized graphene nano-grids as photocatalytic stimulators to accelerate the growth of NSCs into 2D neural networks [50]. Multi-walled carbon nanotubes were unzipped and deposited on a SiO_2_ matrix containing TiO_2_ nanoparticles. After 3 weeks of differentiation, the number of cells increased up to ~26.8-fold. Notably, the nano-grids showed lowered electrical resistance due to the connection by neurite network, implying that formation of cell-to-cell electrical couplings existed in the differentiated network. However, the neural network formed by nano-grids was limited to the 2D plane. Hence, researchers have been searching for substitutes such as nanofiber scaffolds for cultivating neural cells [55]. 

Electrically conductive nanofibers additionally provide cells with favorable 3D architecture and appropriate flexibility similar to neurites in the human body. The attractiveness of using graphene to fabricate nanofibers instead of traditional conductive materials such as carbon nanotubes and poly(3,4-ethylenedioxythiophene) comes from graphene’s superior electrical properties and mechanical strength. While the smooth transition from graphene’s 2D sheet to 3D nanofibers may prove to be difficult, Feng et al. circumvented the problems by assembling a GO sheet onto the surface of electro-spun poly (vinyl chloride) nanofibers, allowing the surface of the nanofiber to be wrapped tightly with graphene [56]. The fibers were covered by strong graphene film of 1–3 nm thickness that showed no exfoliation in ionic environments, and which would be used to form nanofibrous scaffolds with substantially superior size and strength. Mechanical testing showed that the fibers remained completely intact after 5000 twist cycles under a 2.5 circle twist, and had respectable stiffness and maximum conductivity at 6.8 ± 0.67% graphene content. By using this scaffold to culture neural cells with electrical stimulation, acceleration in the proliferation and differentiation of primary motor neurons was achieved. This showed that the graphene nanofibrous scaffold can provide excellent stability, electroactivity, and strength for neural engineering purposes. A similar concept has been applied by Yan et al. to optic nerve recovery with promising results [57]. 

In conditions like glaucoma, high intraocular pressure induces lesions in the optical nerves. While reduction of pressure can halt the damage done to retinal ganglion cells, their function cannot normally be recovered. However, the construction of nanofibrous scaffold using graphene, conductive polymer polypyrrole and polylactic-co-glycolic acid (PLGA) allowed for a high surface area for cell attachment as well as high capacitance for electrical stimulation, guiding cell neurites to regenerate along aligned fiber patterns. Through electrospinning, a fibrous mat structure with diameter at 110 nm was collected. It was shown that the fibers have tunable electroactivity by varying the ratios of graphene and polypyrrole in PLGA, allowing for a flexible therapy for optical nerve-related diseases. Results also showed the importance of electrical stimulation to the anti-aging ability of retinal ganglion cells. Without electrical stimulation, the cells showed decreased size, increased roundness and number of tentacles, resulting in mass apoptosis and lowered cell viability to 40% of the original. The ganglion cells, however, remain healthy after 10 days when electrical stimulation was applied, showing the essentiality of electrical stimulation to neural constructs. Reports have showed that electrical stimulation is essential to calcium-dependent neurite outgrowth and immediate gene expression, and that regeneration rate is frequency-dependent [58]. Hence, it is highly beneficial to control the GN stimulation at higher frequencies at around 10 Hz.

While graphene itself can be used as a substrate, platelets of graphene can also be infused with other materials such as polymers to exhibit specific bulk properties while retaining its conductivity and chemical cues [59]. Although stem cell differentiation for neural regeneration remains at an experimental stage, there has been more success using implanted constructs for healing neurons [60]. Surgically, sutures are a common practice for nerve graft repair in the clinic, at a prolonged regeneration rate of 2–5 mm per day. To improve the healing efficiency, guiding bridges and conduits can be implanted for guiding axon growth in damaged areas. Reports suggest that soft materials containing GNs would be an excellent candidate as implantable materials for neural regeneration due to its tunable stiffness and electrical conductivity [7]. In general, an increase in graphene concentration enhances the crystallization temperature, electrical conductivity, and stiffness of the substrate. For example, Zhu et al. designed graphene bio-ink using gelatin methacrylamide (GelMA) hydrogel, which is renowned for its low stiffness and biocompatibility [61]. The GelMA bio-ink exhibited elastic modulus as low as 30 kPa at 10% GelMA concentration and promoted NSC proliferation and neurite outgrowth in cultured cells after printing well-defined geometry. This shows that the implanting of graphene-infused GelMA bio-ink is a promising approach for nerve repair. For a more clinical approach, Jakus et al. fabricated robust, flexible and electrically conductive 3D printable GNs by mixing graphene with biodegradable PLGA with 20% to 60% solid content [62]. This ratio allowed for the formation of robust and electrically conductive scaffolds comparable to the stiffness of neural tissues. Notably, the elastic modulus of the bio-ink is similar to neural tissues like the spinal cord (1–2.3 MPa). The bio-ink could be utilized to print scaffolds with a diameter as small as 100 μm that support hMSC survival, adhesion, proliferation, and neurogenic differentiation, allowing for elongated morphologies similar to axons and presynaptic terminals. It was shown to be easily handled surgically as it could produce uni- and multi-channel nerve conduits in varying diameters, which naturally break down in vivo (Figure 3). Its safety and flexibility suggest that nerve conduits fabricated from graphene bio-ink are a promising candidate to repair damaged nerves in humans.

Additionally, GNs were applied in neural regeneration through drug loading and delivery through the blood brain barrier, which is normally impassable for drug molecules [63,64]. A nano-scale GO-based delivery system was developed by Dong et al. for treatment of gliomas. GO surface was PEGylated for functionalization of transferrin, allowing for its penetration through the blood brain barrier while the chemotherapy drug Doxorubicin was loaded into the nanocarriers by diffusion and any excess was removed by centrifugation. The system was tested on glioma-bearing mice against injection of freely flowing Doxorubicin, and it showed the mice treated with the GO-based delivery system had a more than double survival time of 40 days. This showed that GO carriers successfully allowed for drug penetration and accumulation inside tumor cells, improving doxorubicin’s efficacy. Functionalization of transferrin also enhanced cellular uptake and targeting. These results suggest that GNs are good candidates as drug vectors for treatment of brain-related diseases. Neural regeneration can be improved by GNs loaded with molecules that aid adhesion such as dopamine (DPA) and arginylglycylaspartic acid (RGD), as evidenced by Qian et al. [65]. They fabricated graphene loaded PCL nanoscaffolds using 3D-printing and a layer-by-layer technique, with much better quality control over electrospinning techniques (Figure 4). Reports suggest that PCL conduits can connect damaged nerve stumps and provide long-term support, while DPA and RGD within the inner tubing improves adhesion. Whilst these molecules dissipate quickly in physiological environments, the GNs utilized controlled release through macropores, allowing for long term implantation and the passage of water and oxygen. The nanoscaffolds were implanted in Sprague Dawley rats and their functional recovery was compared against control. Over 12 weeks, recovery of sciatic nerve was shown to be faster in the Schwann cell-loaded nanoscaffolds compared to other scaffolds without one of the components. This suggests that the synergistic effects of graphene, PCL, adhesion molecules and cells are a promising alternative for peripheral nerve recovery.

### 3.3. Graphene-Based Electrodes for Intracortical Neural Recordings

Neurodegenerative and neurological diseases like Alzheimer’s disease and Parkinson’s disease threaten an ever-increasing percentage of our ageing population. Studies have shown that deep brain stimulation and recovery of sensory capabilities by electrical stimulation based on neural recordings are extremely beneficial to neural regeneration [66]. Hence studying neural signals in the brain is critical for understanding the intrinsic neural pathways in neural regeneration. Graphene-based electrodes are emerging as a novel neural interface due to their flexibility and e conductivity comparable to metals, which are the traditional go-to material for intracortical neural recording [67].

In one study, Kuzum et al. fabricated transparent graphene electrodes for simultaneous electrophysiology recording and neuro imaging using graphene and polyimide [68]. The flexibility and transparency allowed for accurate recording of spatial location and wiring of neurons, which would normally be obscured by metallic parts in traditional electrodes. Neural recording experiments were performed using graphene electrodes and control gold electrodes on an adult rat animal model. The results showed that graphene electrodes had five to six times less noise than gold electrodes, which could be furthered lowered by electrical shielding. Simultaneous calcium imaging and electrical recording was also performed using excitation light at 488nm, and neuronal cell bodies were clearly visible through the graphene electrodes. Graphene layers were shown to provide 100% corrosion resistance to gold electrodes when doped on the surface for up to 6 months in phosphate-buffered saline. These results suggest that transparent graphene electrodes could be a powerful tool in linking low-level neuronal circuits to high-level brain functions. Utilizing similar concepts, Park et al. synthesized carbon-layered graphene for cortical sensing and micro-stimulation to directly stimulate at the electrode-tissue interface [69]. Four graphene monolayers and gold were patterned on silicon to form clear electrodes. While the signals detected were found to be in the same group as platinum electrodes at 95% confidence interval, spatial resolution of the graphene electrodes scales inversely with transparency of graphene due to increased metal parts and traces. Nevertheless, at appropriate proportions, light at 473nm was able to excite mice neurons (Figure 5). Lower intensity lights also caused less spatial spread, activating a relatively focal brain region. These unique properties of the electrodes made possible by their broad-spectrum transparency allow for their broad applications in neural and other biomedical fields.

## 4. Future Challenges

Graphene has been widely proved to play a vitally important role in the proliferation and differentiation of neural cells. Significant progress has been made during the past decades in the development of graphene-based materials [8,55]. Up to now, these preliminary preclinical studies are exciting and encouraging; however, huge challenges still exist and hinder the realization of its future clinical applications.

With the increasing utilization of graphene in a variety of biomedical applications, the in vitro and in vivo biocompatibility of graphene and its derivatives and their probable toxicities must be taken into consideration. The critical parameters which result in toxicity using graphene-based materials are: (1) The size, concentration, and shape. Smaller sizes and higher dose could lead to significant toxicities, giving rise to the fragmentations of DNA and/or chromosomal aberrations in living cells. The potentially sharp edges of graphene flakes or platelets would also result in physical damage of the wall membrane of cells and even nuclei through the direct contact interaction with cells; (2) Their possible aggregation, which may bring about adverse effects (such as generating damaging free radicals) to the blood circulation or immune system [70]; (3) Intracellular/extracellular reactive oxygen species (ROS) generated by the accumulation of graphene-based materials may hinder nutrient uptake, and damage the human body cells and tissues [71]. Therefore, some novel approaches have been adopted to address these problems. For example, the cytotoxicity can be reduced by controlling the size of graphene and the derivatives using synthetic methods. Surface modification including polyethylene glycol (PEG) [72], fetal bovine serum [73] and dextran [74] functionalization has also been utilized to resolve the biocompatibility concerns. However, there is no systematic study addressing the safety concerns of graphene nanomaterials, which is of great importance for future clinical applications in biomedicine.

Additionally, most studies only focus on the exterior cell behaviors upon graphene exposure, thus, the interior effect of intracellular processes should be more emphasized [24,75]. For example, detailed investigations into the cellular uptake mechanism and signaling pathways involved in the progress of stem cell differentiation and network functionality are required to interpret and control graphene-cell interactions. In recent research, Paolo et al. used single layer and multi-layer graphene films to engineer biosensing interfaces, showing a tunable neuronal communication and enhanced ions at the membrane with neurons [76]. Specifically, when the outer stimulations, including the thermal, optical, and electrical ones, are imposed on the graphene nanomaterials, more research is needed to explore the pathology caused by the interaction of the materials and the stimulation [77,78]. Besides, the stimulation of implanted conductive GNFs in vivo should be conducted in animal models for comprehensive understanding and efficacy evaluations in nerve tissue reconstruction.

The possibility of biodegradation of graphene-based materials is another challenging issue, which should be solved especially when graphene impedes the effective interaction of cells and/or exchange of ions [55]. Several reports have demonstrated that the carboxylated derivatives may degrade under appropriate circumstances such as photocatalytic reduction and degradation by TiO_2_ nanoparticles and near-infrared ray-assisted photodegradation to provide a noninvasive method for circumventing this limitation [77,79]. However, the biodegradation may lead to undesired distribution and possible accumulation in blood circulation. Hence, the aforementioned cyto- and genotoxicity analysis of graphene in vivo is highly necessary.

Overall, although there are still various unresolved issues and challenges, GNs are promising substances to pave the way for a real breakthrough in future studies of neural regenerative medicine.

## Figures and Tables

**Figure 1 molecules-24-00658-f001:**
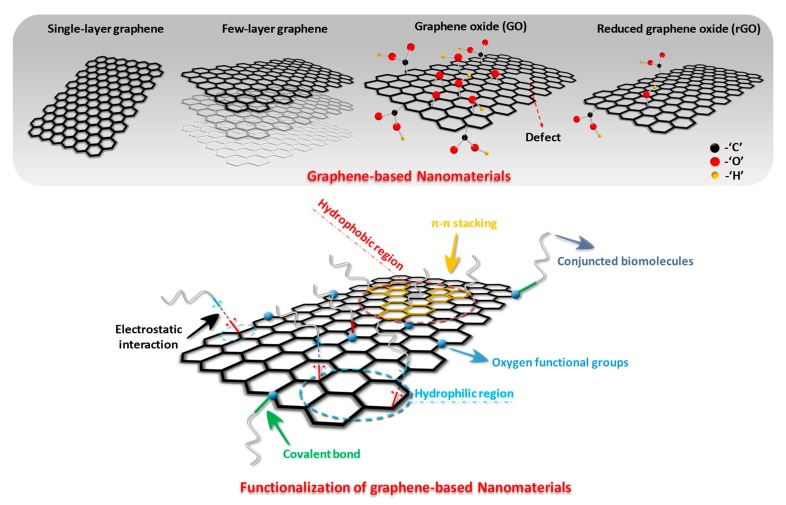
Overview of the various structures of graphene-based nanomaterials (GNs) and the illustration of covalent/non-covalent functionalization of these nanomaterials.

**Figure 2 molecules-24-00658-f002:**
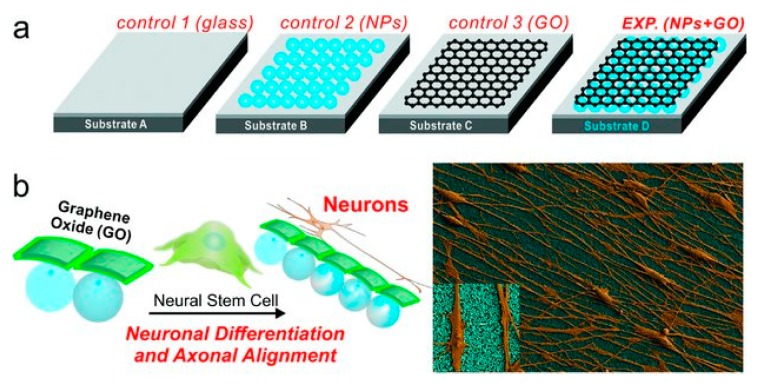
Diagram depicting Solanki’s experimental substrates: glass only; glass with nanoparticles (NPs); glass with graphene oxide (GO); glass with GO and NPs (**a**), and SEM image of aligned axonal growth on GO (**b**). Image retrieved from [51] with permission.

**Figure 3 molecules-24-00658-f003:**
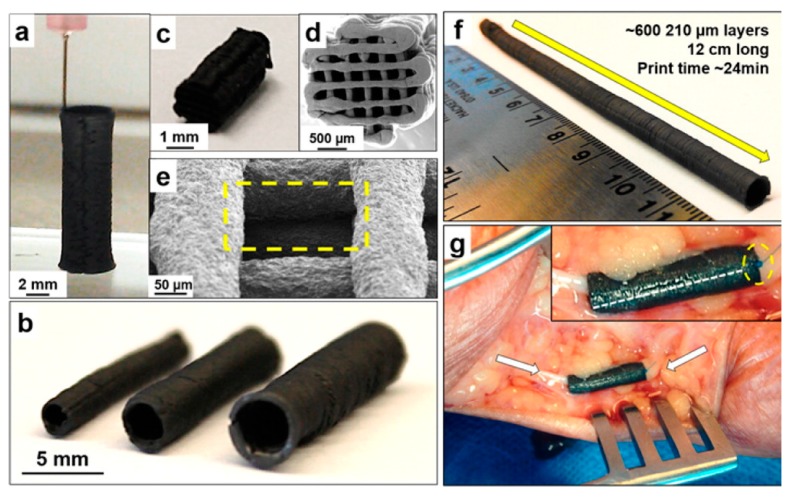
Photographs showing standing strength of 3D graphene ink printed (60% graphene content) nerve conduits (**a**), tubes of variable sizes (**b**), uniaxial multichannel nerve guide (**c**,**d**), SEM micrograph of in-between layers of nerve guide (**e**), a high aspect ratio (24:1) 5 mm diameter hollow tube (**f**), and nerve conduit implanted in human cadaver, demonstrating flexibility and ease of handling of 3D graphene ink (**g**). This shows the flexibility and ease of clinical handling of the graphene ink. Image retrieved from [62] with permission.

**Figure 4 molecules-24-00658-f004:**
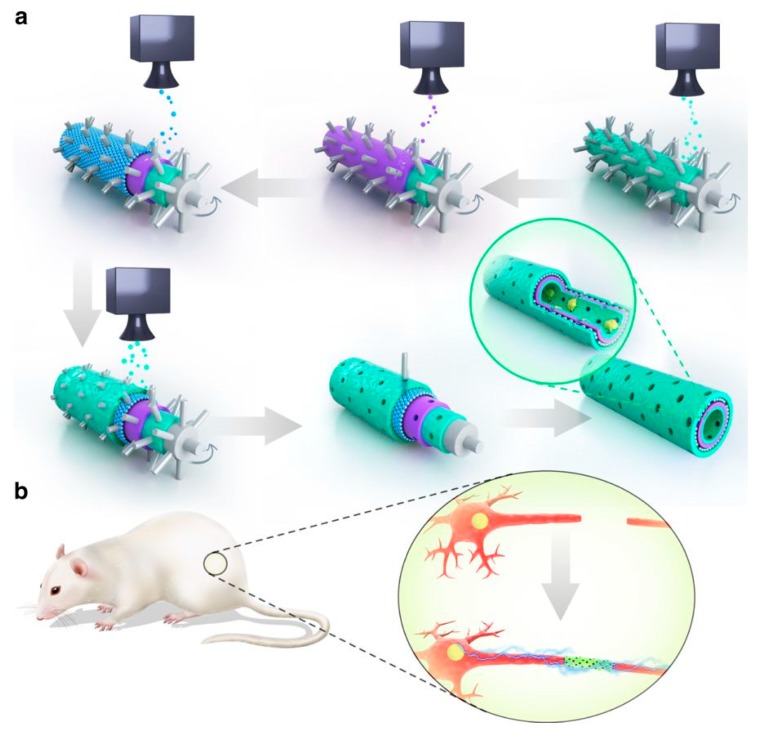
(**a**) Structure and synthesis of GN macroporous nanoscaffold. Green is PDA/RGD, blue and purple are PCL/graphene. (**b**) Illustration of in vivo experimental set up of nerve graft recovery in Sprague Dawley rat. Image retrieved from [65] with permission.

**Figure 5 molecules-24-00658-f005:**
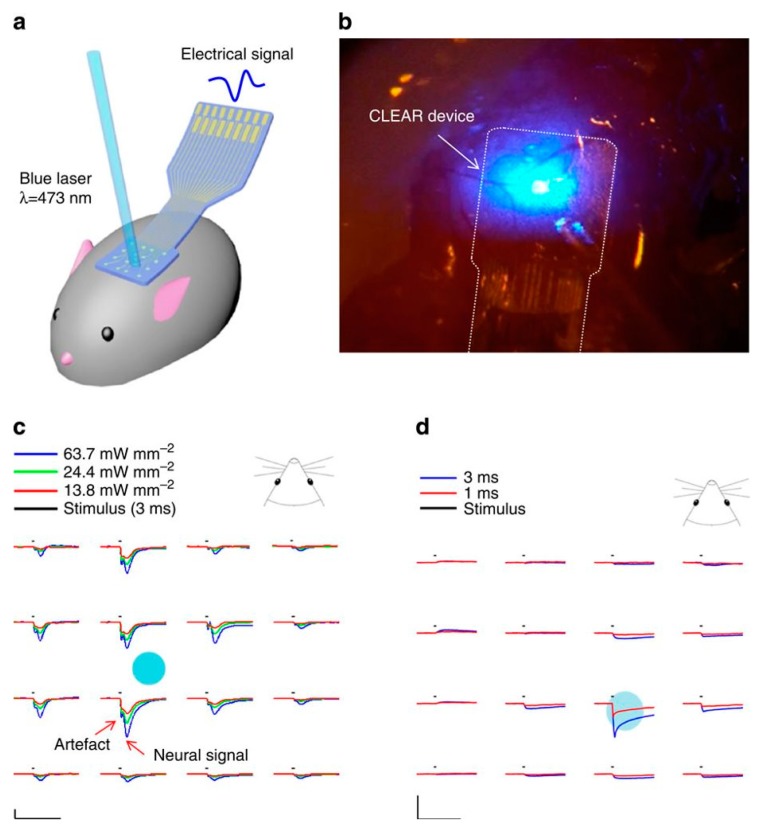
(**a**) Schematics showing the implantation of electrode in mice cerebral cortex. (**b**) Image showing stimulation of cells through electrode using blue light. (**c**) Optical evoked potential data with laser at 24.4 mW mm^−2^. (**d**) Post-mortem control data of same test as (c). Image retrieved from [69] with permission.

**Table 1 molecules-24-00658-t001:** Emerging novel materials in neural tissue engineering and their properties.

Material	Conductivity	Flexibility	Biocompatibility	Reference
**Graphene-based scaffolds**	zero bandgap	flexible	high	[7]
**Indium Tin Oxide**	high	rigid	high	[15]
**Polyaniline**	high	rigid	low	[16]
**Chitosan-based hydrogels**	low	flexible	high	[17]

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
