# Peer review of "Graphene-Based Nanocomposites for Neural Tissue Engineering"

_molecules, 2019, doi:10.3390/molecules24040658_

Reviewer 1 Report

The Review “Graphene-based nanocomposites for neural tissue engineering” by H. P. Bei and co-workers is a nice summary of actual use and possible evolution of graphene-based nano composite materials in the field of cellular/neural tissue engineering.

The manuscript appears well organised and effective in dealing with the proposed topic.

The Reviewer would like just to highlight some minor aspects:

In the abstract the sentence “However, its chemical inertness and potential damage to host body…” is not clear and seems a contradiction. I would suggest to rephrase the sentence.

From the review seems that pristine graphene is not well usable to sustain cellular adhesion and growth… there are many papers where uncoated graphene demonstrated to be able to sustain neuronal cell development (see, for example, Pampaloni et al. Nat. Nanotech 2018). Of course, such substrates are more difficult to be prepared and less versatile than composites. In my opinion a (very) short paragraph highlighting this fact has to be inserted in the review.

Regarding the number of layers of the graphene-based materials: different layers means different properties and different effects on the cells/tissue. Please review this aspect too.

Figure 2 appears having a very low resolution compared to the others figures.

Author Response

Please see uploaded document for response to reviewer

Author Response

Please see uploaded file for response to reviewer

Round  2

Reviewer 2 Report

Authors have adequately addressed my comments. I have no further comments or concerns.